# Parents’ Views with Music Therapy in the Pediatric Intensive Care Unit: A Retrospective Cohort Study

**DOI:** 10.3390/children9070958

**Published:** 2022-06-26

**Authors:** Vladimir L. Cousin, Hubert Colau, Francisca Barcos-Munoz, Peter C. Rimensberger, Angelo Polito

**Affiliations:** Pediatric and Neonatal Intensive Care Unit, Department of Pediatrics, Gynecology and Obstetrics, Geneva University Hospitals, University of Geneva, Rue Willy Donzé 6, 1205 Geneva, Switzerland; hubcolau@googlemail.com (H.C.); francisca.barcos@hcuge.ch (F.B.-M.); peter.rimensberger@hcuge.ch (P.C.R.); angelo.polito@hcuge.ch (A.P.)

**Keywords:** music, PICU, anxiety, quality improvement, questionnaires

## Abstract

Purpose: Music therapy intervention (MT) could be used as an adjunctive therapy in PICU for anxiety and pain management. The aim of the study was to examine the perception of MT by children’s parents in a PICU of a tertiary care teaching hospital. Methods: This is a retrospective cohort study summarizing the results of an institutional quality improvement initiative. Questionnaires were distributed to parents whose children were exposed to MT. Results: From April 2019 to July 2021, 263 patients received a total of 603 h of MT. Twenty-five questionnaires were distributed to parents over a 4-month period (February–June 2021). A total of 19 (76%) parents completed the questionnaire. The majority of parents thought that MT helped their child to communicate (89%), feel less isolated (100%) and cope with stress during hospitalization (100%). The majority of parents also thought that MT contributed to physical recovery (90%) and alleviated feelings of anxiety (90%). Parents also believed that MT should be offered as an out-patient service. Conclusions: Our study agrees with other studies on the positive potentials of MT in PICU. Music therapy intervention could be used to promote children’s and parents’ psychological well-being. Further studies are warranted to evaluate the impact of MT on long-term post-ICU outcomes.

## 1. Introduction

The main goal of the use of music in intensive care units (ICU) is to reduce anxiety and pain [1]. Music in ICU has been shown to reduce respiratory rate, blood pressure and heart rate [2]. Music interventions are also associated with decreased pain, sedation and post-traumatic stress disorder in children operated on for congenital heart disease [3].

According to the American Music Therapy Association, music therapy intervention (MT) is “the clinical and evidence-based use of music interventions to accomplish individualized goals within a therapeutic relationship by a credentialed professional who completed and approved music therapy program” (available online: https://www.musictherapy.org/about/musictherapy/ (accessed on 14 June 2022)). Music therapy intervention has the potential to alleviate pain, promote physical rehabilitation and help manage stress. Its use is recommended by the Society of Critical Care Medicine in both pediatric and adult ICU [4,5]. The use of MT in neonatal ICUs also suggests that MT is feasible and might have a positive impact on premature children. Little has been reported about the effect of MT in the pediatric ICU (PICU) setting and especially how parents respond to MT [6,7,8,9]. Therefore, the aim of this study was to describe the perception of MT by parents of children who were exposed to MT in PICU.

## 2. Methods

This quality improvement initiative was supported by the Foundation Alta Mane. It was implemented in April 2019 in a tertiary 12-bed PICU at the Geneva Children’s Hospital, Switzerland, and is still running as of today. As a retrospective study on a quality improvement project, and in agreement with institutional policy, this study was exempt from ethics committee review. Nonetheless, parents’ oral consent was required for participation in the study.

We aimed to evaluate the perception of such an initiative by the parents of a sample of children admitted to our unit. An evaluation of the program was performed by means of two five-point Likert scale questionnaires. The questionnaires were designed and created by the authors to collect views from parents whose children received music intervention in the PICU. This questionnaire was adapted from Moss et al. [10,11], but without previous validation. It was addressed to parents whose children were exposed to MT throughout a five-month survey period (February–June 2021, Appendix A). Only patients who were fully responsive and not receiving sedation were exposed to MT. A personalized approach to MT was used whenever possible. In order to avoid possible MT side effects (e.g., anxiety, overstimulation and unwanted memory triggering), MT was tailored to the patients’ needs and constantly adapted to the child’s responses to music and sound. Two versions of the questionnaire (French and English translation) were given to parents. Participants were encouraged to add comments and suggestions at the end of each questionnaire.

Hospitalized patients participated in personalized MT with a certified music therapist (HC). The therapist was present 4 days per week, for a total of 15 h per week. Music therapy interventions were provided at the bedside in each patient’s room. The duration of sessions varied between 25 and 60 min based on the patient’s needs, preferences and clinical conditions. Parent’s and child’s direct participation are considered key components of MT in our PICU, and were encouraged whenever possible. No psychotherapy was offered during MT. Several different instruments were used during MT sessions (kalimba, rain stick, guitar, ukulele, metallophone, small djembe and other percussions).

Descriptive analyses were performed by VLC and AP, using Microsoft Excel 2013 (Microsoft Corporation, Redmond, WA, USA). As our study is solely descriptive, no statistical comparisons were performed.

## 3. Results

During the study period, 263 patients received a total of 603 h of interventions. The main primary diagnoses were represented by congenital heart diseases (155 patients, 59%), respiratory diseases (42 patients, 16%), central nervous system diseases (24 patients, 9%), post-surgical interventions (16 patients, 6%) and solid organ transplantations (8 patients, 3%).

Parents’ views on MT were studied over a 4-month period (February–June 2021); a total of 25 questionnaires were handed over to the parents of PICU patients whose mean age was 7 years (SD+/ −6). The number of sessions per patient was 1 to 2, 2 to 4 or ≥5 in 50%, 39% and 11% of patients, respectively.

One parent per patient answered the questionnaire, with a response rate of 76% (19/25). The results from the parents’ questionnaires are summarized in Figure 1. The main reasons for PICU admissions of children whose parents responded to the questionnaire were congenital heart diseases (3/19), central nervous system diseases (3/19), liver insufficiency (3/19) and respiratory insufficiency (3/19). All parents thought that MT was beneficial for their child. Specific benefits of MT, as stated by parents, included helping to communicate (89%), feel less isolated (100%) and cope with being in hospital (100%). Overall, 100% parents felt that their child was respected and supported in MT. The majority of parents also thought that MT contributed to physical recovery (90%) and alleviated feelings of anxiety (90%). Of the parents surveyed, 63% thought that MT facilitated communication with the PICU team. The majority of parents (93%) also believed that MT should be offered as an out-patient service. A few examples of open-ended responses are shown in Figure 1.

## 4. Discussion

In this single-center study, we showed that MT could introduce alternative ways of supporting children’s and parents’ psychological and possibly physical well-being. We also found that MT might promote constructive communication between families and the PICU team.

Our results are in accordance with recent literature showing that MT is feasible and is accompanied by a high level of appreciation from staff and families in the PICU setting [12,13]. Music therapy interventions might also reduce anxiety and pain in critically ill children [14,15,16]. Our results underlined the complex role of MT with a potential impact on multiple aspects of patients’ and parents’ perceptions during their stay in the PICU. Actually, the Society of Critical Care Medicine strongly recommended MT as part of a multicomponent intervention aimed at reducing analgesic use and improving pain management in pediatric patients in ICU [5].

Our report has several limitations. Our results are based on a small sample size. Although it is based on an already published questionnaire, the survey used in this study was created by the authors for the sake of this analysis and has never been validated. We chose the survey approach as our experience with the qualitative interview approach is limited. We believed that, although less informative, the use of an already published survey might have helped us to overcome the lack of familiarity with the qualitative interview approach. However, similarly to humanities in general, MT is a complex phenomenon and is more than just listening to music [17]. As such, the real impact of MT on patients and parents might be only partially captured by objective measurable components [18]. We were able to confirm the usefulness and feasibility of MT in the PICU in terms of patient-centered outcomes and the team’s acceptance. Further research looking at patients’ and their parents’ experiences is warranted to accurately adapt MT interventions to patients’ needs and help promote children’s autonomy.

## Figures and Tables

**Figure 1 children-09-00958-f001:**
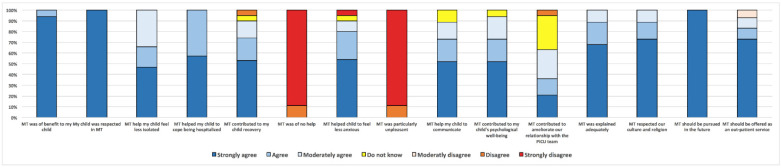
Parents’ answers to questionnaire. *Y* axis: proportion in %. MT: music therapy.

## Data Availability

Data are accessible upon reasonable request to the corresponding author.

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
