# Peer review of "Parents’ Views with Music Therapy in the Pediatric Intensive Care Unit: A Retrospective Cohort Study"

_children, 2022, doi:10.3390/children9070958_

Round 1
Reviewer 1 Report
Review - Children
This article may be essential and it might contribute to the implementation of MT services in health institutions. However, I am concerned about the study's ethics and methodology parts, which are poorly presented in the article.
A few comments:
The title should reflect the article better - I would suggest to change:”experience” (implies phenomenology) with “views”, and adding the method - retrospective cohort study
Intro
The first reference (Sliwka et al) on what MT is an indirect reference, please find a direct reference to MT
lines - 32-36
Please be more precise on the difference between music interventions and Music Therapy. These are not the same and there is constant confusion between them throughout the manuscript.
F.E: References number 4 - 6 are music-based interventions and not MT, however, are presented as MT outcomes in the intro.
Method
The Method part is clearly non-academic. Many issues are unclear or missing. Although not suited for ethical commits, I presume a consent form is required…
More examples:
Examples of the questionnaire should be given to the reader
There is no mention of how data was analysed, who analysed etc.
There is no reference or specification of intervention protocol.
Line 58 - sentence not clear, it seems that each participant of MT receives 15 hours of care?
Line 52-56 - is not clear and maybe misleading the reader in thinking that you have selected only patients who the therapist thought benefited from MT?
Author Response
We thank the editor and the reviewers for their comments that we have taken into account in order to further improve the quality of the manuscript.
We are pleased to submit a revised version of the manuscript entitled « Parents’ view on music therapy in the Pediatric Intensive Care Unit : a prospective cohort study ». We hope that our manuscript will be of interest for the readers of Children.
REVIEWER 1
This article may be essential and it might contribute to the implementation of MT services in health institutions.
However, I am concerned about the study's ethics and methodology parts, which are poorly presented in the article.
A few comments:
The title should reflect the article better - I would suggest to change:”experience” (implies phenomenology) with “views”, and adding the method - retrospective cohort study
R : Thank you. The title was changed as suggested.
We add in the method a better description of the design of the study. It was a retrospective cohort study.
Intro
The first reference (Sliwka et al) on what MT is an indirect reference, please find a direct reference to MT
R : Thank you for your comment. The first sentence was rephrased and a direct reference to Mt was added.
“MT as a controlled form of listening to music and its influence on the person, physiologically, psychologically, and emotionally, during treatment of illness or injury “. (Biley et al, Nursing standard (royal College of Nursing (Great Britain) 1992).
lines - 32-36
Please be more precise on the difference between music interventions and Music Therapy. These are not the same and there is constant confusion between them throughout the manuscript.
R: We agree with the reviewer. „Music intervention“ was substituted with „music therapy intervention“ throughout the manuscript.
F.E: References number 4 - 6 are music-based interventions and not MT, however, are presented as MT outcomes in the intro.
R : Correct, thank you. The introduction has been rephrased as follows :
“ The main goal of the use of music in ICU is to reduce the anxiety and pain of patients [4]. Music in ICUs has been shown to reduce respiratory rate, blood pressure and heart rate [5]. Music interventions are also associated with decreased pain, reduced sedation and post-traumatic stress disorder in children operated on for congenital heart disease [6].”
Method
The Method part is clearly non-academic. Many issues are unclear or missing. Although not suited for ethical commits, I presume a consent form is required…
We thank the reviewer for having raised this important issue. Actually we should have given more details on that.
In our institution, quality improvement projects are considered low-risk research and are exempt from ethics reviews, as it is the case in several countries (Scott AM et al, health research Policy and Systems, 2020). This project is a quality improvement project aimed at improving patients’ support practices in our unit. Only parents that orally agreed to fill in the questionnaire actually did it. Therefore no consent to participate was collected.
More examples:
Examples of the questionnaire should be given to the reader
R : Thank you. An example of the questionnaire is available in the supplementary material.
There is no mention of how data was analysed, who analysed etc.
R : More details regarding the analysis have been added to the manuscript (line 82-83, methods section).
There is no reference or specification of intervention protocol.
R: We thank the reviewer for this comment. We favor a striclty personalized approach to MT for our patient. Therefore intervention protocols were not available during the study period. Nonetheless a description of MT intervention is available in the ‚Methods’ section (page 2, line 59-63 and line 68-81).
Line 58 - sentence not clear, it seems that each participant of MT receives 15 hours of care?
R : The aforementioned sentence was amended. The therapist is present 4 days per week, for a total of 15 hours.
Line 52-56 - is not clear and maybe misleading the reader in thinking that you have selected only patients who the therapist thought benefited from MT?
R : Thank you for this comment. The sentence was rephrased as follows :
« Only patients who were fully responsive and not receiving sedation participated in the MT sessions. In order to avoid possible MT side effects (i.e. anxiety, overstimulation, unwanted memory triggering) only patients whose emotional responses to music might be evaluated by the music therapist actually benefited from MT ».
Reviewer 2 Report
This short report aimed to describe parental experiences with music therapy in a pediatric intensive care unit (PICU) setting. The authors used a non-validated questionnaire to capture parental experiences with music therapy. Responses by parents showed that they valued music therapy for their child’s benefit, and it ameliorated relationships with the PICU team.
I have the following comments and questions:
(1) The introduction is well written and provides the gap in the evidence. The introduction could be improved by including a couple of sentences on music therapy in the NICU, as several of the principles used there may be applicable in PICU settings as well.
(2) What do the authors mean by the sentence “Only patients who could participate in the MT sessions and whose emotional responses to music might be evaluated by the music therapist actually benefited from MT.” (Lines 52-54)? Does that mean that patients had to be fully responsive and could not receive sedative medication?
(3) Can the authors elaborate on the session goals that were addressed by the music therapist?
(4) Were the music therapy sessions individual or with parent(s) present at bedside, and If so, was psychotherapy included in the sessions?
(5) Can the authors explain why they chose a survey approach as opposed to a qualitative interview approach to further capture actual experience rather than attitudes towards music therapy?
(6) Would the authors consider adding pediatric patients’ experience as a future direction of research to their Discussion section? In my opinion, the child should also have a say in their care and this intervention is a great opportunity to provide the child with autonomy in an otherwise strict setting.
Author Response
We thank the editor and the reviewers for their comments that we have taken into account in order to further improve the quality of the manuscript.
We are pleased to submit a revised version of the manuscript entitled « Parents’ view on music therapy in the Pediatric Intensive Care Unit : a prospective cohort study ». We hope that our manuscript will be of interest for the readers of Children.
This short report aimed to describe parental experiences with music therapy in a pediatric intensive care unit (PICU) setting. The authors used a non-validated questionnaire to capture parental experiences with music therapy.
Responses by parents showed that they valued music therapy for their child’s benefit, and it ameliorated relationships with the PICU team.
I have the following comments and questions:
1_The introduction is well written and provides the gap in the evidence. The introduction could be improved by including a couple of sentences on music therapy in the NICU, as several of the principles used there may be applicable in PICU settings as well.
R: We agree with the reviewer. We add more information on the use of music therapy in NICU.
Further references were also added.
2_What do the authors mean by the sentence “Only patients who could participate in the MT sessions and whose emotional responses to music might be evaluated by the music therapist actually benefited from MT.” (Lines 52-54)? Does that mean that patients had to be fully responsive and could not receive sedative medication?
R : Thank you for this comment. The sentence was rephrased as follows :
« Only patients who were fully responsive and not receiving sedation participated in the MT sessions. In order to avoid possible MT side effects (i.e. anxiety, overstimulation, unwanted memory triggering) only patients whose emotional responses to music might be evaluated by the music therapist actually benefited from MT ».
3_Can the authors elaborate on the session goals that were addressed by the music therapist?
R : the following sentence was added to the „Methods“section:
« Music therapy interventions aimed at creating a protected and comforting environment that could alleviate pain and anxiety, manage stress and promote physical rehabilitation by means of, whenever possible, a direct participation of the child.».
4_Were the music therapy sessions individual or with parent(s) present at bedside, and If so, was psychotherapy included in the sessions?
Music session were performed with the parent(s) present at bedside whenever possible. No psychotherapy was offered during MT interventions.
5_Can the authors explain why they chose a survey approach as opposed to a qualitative interview approach to further capture actual experience rather than attitudes towards music therapy?
R : The reviewer is correct. This represent a limitation of our study that we acknowledged in the manuscript. We chose the survey approach as our experience with qualitative interview approach is limited. We believed therefore that, although less informative, the use of an already published survey might have helped us overcome the lack of familiarity with qualitative interviews.
6_Would the authors consider adding pediatric patients’ experience as a future direction of research to their Discussion section? In my opinion, the child should also have a say in their care and this intervention is a great opportunity to provide the child with autonomy in an otherwise strict setting.
R : We completely agree with the reviewer. The following sentence was added to the discussion section :
« Future research looking at patients’ and their parents experience is warranted to accurately adapt MT interventions to patients’ needs and help promote children’s autonomy ».
Round 2
Reviewer 1 Report
Dear Authors,
I appreciate your detailed answers and changes to the manuscript.
I do however feel that some parts of the article are unclear or not precise and need to be reworked. I also advise performing English editing.
Please see my following comments:
Title - parents’ views should be in plural
Abstract
the abstract should go under major English editing. F.E: “Children parents” should be "children’s parents"; perception should be changed to perceptions etc.
line 22 of the abstract - “our study confirmed” - confirmed is a too strong word for such a small scale study, consider changing it to “our study agrees with other studies on positive potentials of MT” or other wording that is less generalising.
Introduction
The new reference on defining MT is still not related to music therapy but to musical uses In the ICU and is a very old reference…MT is not only about listening to music as you state in your discussion…
The introduction lacks an explanation of what MT is. This should be added after the section on music interventions.
Method
Line 62 “only patients whose emotional responses to music might be evaluated by the therapist“ - is still unclear. What does it mean that a therapist can evaluate emotional responses to music? And if so, how did the therapist make this evaluation? The reader needs to understand this to increase the trustworthiness of the study.
Line 73 - MT intervention - I appreciate the elaboration on the type of MT interventions used. Was the parent or child (when possible) encouraged to participate? Was it a passive way of client-only listening to the music played by the music therapist? Please elaborate.
Author Response
We are grateful to the Editor and the Reviewers for their comments that we have taken into account in order to further improve the quality of the manuscript.
We are pleased to submit a revised version of the manuscript entitled « Parents’ view on music therapy in the Pediatric Intensive Care Unit : a retrospective cohort study ». We hope that our manuscript will be of interest for the readers of Children.
Dear Authors,
I appreciate your detailed answers and changes to the manuscript.
I do however feel that some parts of the article are unclear or not precise and need to be reworked. I also advise performing English editing.
Please see my following comments:
Title - parents’ views should be in plural
R: thank you. That was done.
Abstract
the abstract should go under major English editing. F.E: “Children parents” should be "children’s parents"; perception should be changed to perceptions etc.
R: Thank you. The entire manuscript underwent a thorough English revision. We hope that it reads better now.
line 22 of the abstract - “our study confirmed” - confirmed is a too strong word for such a small scale study, consider changing it to “our study agrees with other studies on positive potentials of MT” or other wording that is less generalising.
R: We thank the reviewer. We rephrased the sentence as suggested.
Introduction
The new reference on defining MT is still not related to music therapy but to musical uses In the ICU and is a very old reference…MT is not only about listening to music as you state in your discussion…
The introduction lacks an explanation of what MT is. This should be added after the section on music interventions.
R: We agree with the Reviewer. We added the AMTA definition of MT with the relative reference to the introduction after the section on music intervention. Some rephrasing was also added throughout the introduction section.
Method
Line 62 “only patients whose emotional responses to music might be evaluated by the therapist“ - is still unclear. What does it mean that a therapist can evaluate emotional responses to music? And if so, how did the therapist make this evaluation? The reader needs to understand this to increase the trustworthiness of the study.
R: We agree. The aforementioned sentence is not clear and was therefore rephrased as follows:
“A personalized approach to MT was used whenever possible. In order to avoid possible MT side effects (i.e. anxiety, overstimulation, unwanted memory triggering) MT were tailored to patients’ needs and constantly adapted to child’s responses to music and sound ».
Line 73 - MT intervention - I appreciate the elaboration on the type of MT interventions used. Was the parent or child (when possible) encouraged to participate? Was it a passive way of client-only listening to the music played by the music therapist? Please elaborate.
R : Thank you: The following sentence was added to the methods section:
”Parents’ and child’s direct participation are considered key components of MT in our PICU and were encouraged whenever possible. No psychotherapy was offered during MT.